# The Struggle for Human Attention: Between the Abuse of Social Media and Digital Wellbeing

**DOI:** 10.3390/healthcare8040497

**Published:** 2020-11-19

**Authors:** Santiago Giraldo-Luque, Pedro Nicolás Aldana Afanador, Cristina Fernández-Rovira

**Affiliations:** 1Communication and Journalism Department, Autonomous University of Barcelona, 08193 Barcelona, Spain; santiago.giraldo@uab.cat (S.G.-L.); pedronicolas.aldana@e-campus.uab.cat (P.N.A.A.); 2Communication Department, University of Vic-Central University of Catalonia, 08500 Barcelona, Spain

**Keywords:** attention, social media, digital wellbeing, neuropsychology, neurophysiology, economy, addictive design

## Abstract

Human attention has become an object of study that defines both the design of interfaces and the production of emotions in a digital economy ecosystem. Guided by the control of users’ attention, the consumption figures for digital environments, mainly social media, show that addictive use is associated with multiple psychological, social, and physical development problems. The study presented develops a theoretical proposal regarding attention. In the first part, the research analyzes how attention has been studied and how it behaves using three disciplines: neurophysiology, neuropsychology, and economics. In the second part, considering this general framework, the study uses categories of the three disciplines to explain the functioning of social media, with special emphasis on their interactive, attractive, and addictive design. Finally, the article presents, as a practical example of the exposed theory, the main results of two case studies that describe social media consumption among young people. The research shows the relevance of the theoretical study of attention as a key element by which to understand the logics that dominate the interactive design of social media. It also uses a multidisciplinary perspective. The addictive behaviors identified in the two examples support the theoretical proposals and open research lines oriented to the measurement and understanding of the attention given to social media.

## 1. Introduction

The concentration of attention on social media determines a field of analysis oriented to the understanding of consumption focused on digital attention. This line of research also includes strategies that large digital platforms use to capture users’ attention and data, two of the most valuable goods in digitized society. The capacity of the interfaces to generate interactions and clicks is crucial in the current economic system in which the object of study is inserted. It is important to discover how this addictive design works. This understanding constitutes a social need nowadays, when we see how the digital behavior of more than 5 billion users of social media can be monitored [1]. Moreover, digital attention is concentrated in only five companies: Facebook Inc (California, United States), Alphabet (California, United States), Tencent (Shenzhen, China), ByteDance (Beijing, China), and Sina Corporation (Beijing, China).

Attention, understood as the time of consumption, is valuable data, as it serves to predict and guide future behavior. Therefore, capturing the user’s attention is the main objective of digital platforms and defines the field of action of the economy of attention. It is “an economic project that reflects the mutation of a new capitalism” [2] centered on the domination and control of the cognitive production of the platform users and which can be defined as the management of the scarce resource of human attention.

The attention economy outlook calls for the need to approach the dynamics of human attention study. Thus, this article presents the attention research development, followed by a conceptual approach to attention using three disciplines: neurophysiology, neuropsychology, and economy. It also describes how the disciplinary elements presented are associated with the design and consumption of social media. Finally, the article shows the results of two case studies to illustrate the effectiveness of the mechanisms used by social media to capture human attention, as a complement to the theoretical approach.

According to Filley, “the representation of attention in the brain is thus widespread, consistent with its essential role in human mental life” [3]. Recent studies have shown that young people spend an average of 5.5 h per day connected to social media [4]. This is almost a third of the daily active hours of any person. If attention is on social media, it is important to understand why.

### 1.1. The Attention Research Development

Epistemologically, attention is a multidisciplinary term that groups together diverse processes. To clarify what human attention is and how it works, we start by understanding the research development of the concept. Then, we observe it from the point of view of neurophysiology, neuropsychology, and economics to reveal its operation within the individual emotional and sensory system, as well as its development within social science.

#### 1.1.1. Brief History of the Attention Research Development

Attention has been studied by fields such as optics, biology, neurophysiology, neurosciences, psychology, communication, pedagogy, and economics. The attention research was born in the 19th century, but “before this time, philosophers had typically considered attention within the context of apperception (the mechanism by which new ideas became associated with existing ideas)” [5]. One of the first researchers to use the concept was Gottfried Wilhelm Leibniz, who “suggested that attention determines what will and will not be apperceived” [5].

Psychology developed a study of attention that can be presented in three stages. The first phase, promoted by James, Helmholtz, Müller, Pillsbury, Tichener, and Wundt, applied an introspective method of the human mind. Wundt, one of the founding fathers of structuralism, “wrote of the wide field of awareness (which he called the Blickfeld) within which lay the more limited focus of attention (the Blickpunkt) (…) He also speculated that attention is a function of the frontal lobes of the brain” [5]. At the same time, William James [6] characterized attention from perception, distinction, and remembrance, highlighting its selective function as well as the motivation and interest associated with it. James argued that the individual only becomes aware of the stimuli that are attractive to him.

Hermann von Helmholtz’s experiments announced that an “observer who is steadily gazing at a fixation mark can, at the same time, concentrate attention on any given part of the visual field” [7]. An alternative pioneering study by Titchener [8] conceptualized attention as an attribute that makes the contents reach greater clarity in consciousness. Thus, the authors of this first phase state that attention is important, although it is a limited capacity.

The second stage brings together the research carried out related to attention by the Behavioral and Gestalt Schools. For these schools, the first studies lacked methodical and experimental elements, which called into question their objectivity. According to Watson and Skinner, the concept of attention was unscientific, and they debated whether it should enter the psychology studies [9]. However, for Berlyne [10] attention remained an object of study in the field and was characterized as alertness. Therefore, attention was not only limited to a selection but was linked to alertness. Moreover, Berlyne proposed that alerting depended on the form and intensity of the stimulus, but at the same time on its collative properties: complexity, novelty, incongruity, and surprise [10], relationships that link attention to the Ascending Reticular Activation System (ARAS).

Thanks to the reflexology studies of Pavlov [11], Bechterev [12], and Sechenov [13], attention is considered a behavior similar to the orienting reflex or response. It is a physiological behavior that leads to analyzing the individual response to different stimuli and to observing the electrophysiological, vascular, and motor changes. Razran [14] defined it as the first organic reaction to a stimulus that generates a change. Thus, the second stage shows the relevance of studying the composition of the received stimuli for their selection and effectiveness and the attention as a gateway to human behavior.

The third phase of research on attention is formed by cognitive psychology and the contribution of neurosciences. As a result of information processing theory, as formulated by Claude Shannon [15], academic efforts are oriented towards understanding the process of cognition. In this way, the first information processing models in which attention is considered an information filtering mechanism are consolidated: Broadbent’s filter model [16], Treisman’s Attenuation model [17], the Deutsch and Deutsch model [18], and Norman’s model [19]. Subsequently, other models that consider attention as in charge of distribution and carrying out different tasks are presented: Kahneman’s model [20] and the Norman and Bobrow model [21]. The last stage studies the relationship of the visual field in a particular way; it investigates the brain’s response with more specific and effective forms of measurement and explores the divided function of attention.

The stages propose a classification of research into three major fields. The first one is selective attention, which identifies attention as a process of the selection of stimuli. The second one is divided attention, which studies the response capacity of the subject to several simultaneous tasks. The third one is sustained attention, which examines the ability to retain attention for an amount of time, which is called concentration.

The attention research development is broad and diverse, due to the multiple implications of the processes involved in it. The most advanced techniques for studying and monitoring brain activity make it possible to better understand the physiological functioning of attention and how it relates to other executive functions of the human being.

#### 1.1.2. Attention in the Field of Neurophysiology

Neurophysiology defines attention as the ability to focus selectively on an object or task, and it is essential for a series of actions of the brain that link different cognitive functions. Similarly, it is understood as an activity increase in a certain brain area involved in the processing of a stimulus. Some researchers propose that attention is “the interface between the vast amount of stimulation provided by our complex environment and the more limited set of information of which we are aware” [22], taking the perspective of a “selection machine”. Additionally, “attention has been largely linked to the voluntary and effortful control of action” [22], which determines that the term leads to the “generation of voluntary behavior” in the individual, relating it to the concept of arousal (brain activation that includes the rhythm of brain processes) and states of consciousness.

“Researchers in the field agree that attention is not a unitary term. Rather, we can fractionate attention into subsystems of more circumscribed function and anatomy” [23]. Thus, the neuroscientific understanding of attention is chosen from the integrative Attention System theory proposed by Posner and collaborators [24,25,26,27]. “This theory states that the variety of attentional manifestations is produced by separate but related attentional systems” [27]. The model is formed by the alerting network, the posterior attention network (orienting response), and the anterior attention network (executive function).

The first network, called alerting, “involves a change in the internal state in preparation for perceiving a stimulus. The alert state is critical for optimal performance in tasks involving higher cognitive functions” [23]. It determines the changes in the individual state of consciousness and allows a direct link to the arousal in order to be traced. The neuroanatomical functioning of the first network is located at the locus coeruleus and in the right frontal and parietal cortex [23,28,29,30,31]. The presence of norepinephrine, which acts as a neuromodulator, is also necessary [32,33,34,35].

The second one is called the orienting network and “concerns the selection of information from a sensory input” [23]. The orienting network helps to prioritize sensory information and leads individuals to keep their focus on the object or action they perform. The network discriminates the relevance of the stimulus and provides the skill to maintain interest in what has captured the subject’s attention. Likewise, it allows splitting the attention into two or more activities when necessary, although one stimulus will always be predominant.

From the neuroanatomical point of view, the orientation network is mainly found in the superior parietal lobe, in the superior temporoparietal junction [36], in the superior colliculus, and in the frontal ocular fields [37,38], and its predominant neuromodulator is acetylcholine. As is anatomically evident, in the second network the visual element predominates since the selection of the stimuli evidences the Theory of Biased Competition [39], which “sees attention as arising out of a winner-take-all competition within various levels of sensory and association systems” [40].

Finally, the last network is called the executive. Through it, the neural system sharpens the brain to focus attention on the object or action performed, which limits the ability to react to other stimuli. This function leads to concentration or the ability to sustain attention for a length of time. “Executive control of attention involves more complex mental operations both to monitor and resolve conflicts between computations occurring in different brain areas. Executive control is most needed in situations involving planning or decision-making, error detections, novel or not well-learned responses, difficult or dangerous conditions, and in overcoming habitual actions” [23]. The neuroanatomical location of the third network is the Anterior Cingulate Cortex [41,42], the lateral ventral prefrontal cortex, and the basal ganglia. The predominant neuromodulator of the executive network is dopamine.

Attentional networks have two processing mechanisms. The first one, top-down, “represents the selection processes intended for particular goals, which produces greater neuronal activation of the relevant sensory input to discriminate the stimulus of interest from those not relevant in order to achieve the goal” [43]. The second, bottom-up, “is associated with the processes that take action when attention is directed to a particular stimulus because certain characteristics of the stimulus excel, such as its infrequency, novelty, intensity or contextual relevance” [43].

The brain contains complexities and attention is one of them. “At its most fundamental level, attention is represented in the human brain as a widespread collection of interconnected structures that has been called the attentional matrix” [3]. Many challenges concerning executive functions are related to issues of attention. Hence, attention is “a complex neurobehavioral capacity without which the expression of all other higher functions of the human brain is impossible” [3].

#### 1.1.3. Attention in the Field of Neuropsychology

Neuropsychological studies of attention endorse its complexity. According to Ribot, it is difficult “to distinguish where it begins and where it ends” [44]. Styles, in turn, affirms that “attention is a term that comprehends diverse psychological phenomena” [45]. Despite its complexity, Tudela proposes a conceptual approach to attention as “a central mechanism of limited capacity whose primary function is to control and guide the conscious activity in accordance with a specific objective” [46]. García Sevilla extends this idea, proposing that attention is “a mechanism that launches a series of processes or operations thanks to which (…) we are more receptive to events in the environment and allows us to perform numerous tasks more efficiently” [47].

In the same way, García Sevilla states that there are three types of processes involved in the attentional mechanism: selection, distribution, and sustaining [47]. Selection is the most common of them and allows one to respond to a specific stimulus in the environment even though there are more of them. Bonnet [48], Broadbent [16], and James [6] articulate attention from this selective perspective. Furthermore, the distribution process helps the individual to respond to multiple stimuli, an approach to what today is known as multitasking. This is proposed by Boujon and Quaireau [49], in addition to Sternberg [50], who researches the development of different tasks at the same time, together with Block [51] and Burt and Kemp [52], who suggest the Time Estimation Paradigm, which consists of exploring the expected time for a specific task.

The last process is sustaining, which leads to focused attention, commonly known as concentration. This process is responsible for maintaining as long as possible the attention to the stimulus assigned by the subject. Some authors have linked this concept with cognitive psychology and learning, as illustrated by Rabiner and Coie [53]. Willcutt and Pennington [54] have linked it with the process of learning to read, Valencia and Andrade [55] with behavior control, Berthiaume [56] with reading comprehension, and Barkley and Murphy [57] with the study of problem-solving.

Additionally, neuropsychology proposes three moments to describe the functioning of attention: the interest (attention capture), sustaining, and the finalization of the attentional process. The first two are similar to the functions of the attentional networks described before (selection, distribution, and sustaining). However, as a possible cause of the termination of the attentional activity, the subject may present fatigue or tiredness due to the action performed. Additionally, the finalization of the attention may be produced by the fact that the activity is monotonous or creates boredom. Nevertheless, there may be a spontaneous recovery [47] when, despite the sustained attention loss, the neuropsychological system refocuses on a stimulus [11,58,59].

Broadbent [16], Treisman [17], and Styles [45] analyze the three moments of attention and their multisystemic characteristics, as well as defining attention as a limited capacity. Although the individual wants to attend to all the stimuli received, it is imperative to focus on one of them. Regardless of how much the function is divided, the response and efficacy shall not be the same [60]. This is one of the fundamental principles of attention economy.

Another relevant factor is the stimuli magnitude, which is proportional to the individual response [16,61,62]. At the same time, if a stimulus is new, it generates a greater impact or intensity than a repetitive one. This is a phenomenon called habituation, and it is formed by the frequency with which the same stimulus is received and the rate of its appearance, which generates a loss of interest in the subject and a decrease in neural sensitivity to its response.

Attention might be overwhelmed by the channel’s oversaturation of stimuli. This causes, on the one hand, the subject to not focus on the main stimuli of interest, causing a loss of information due to inattention or dispersion. On the other hand, it can provoke distraction, leading to the termination of the attentional function.

Attention plays a fundamental role as a selective action of environmental stimuli, which makes neurophysiology and neuropsychology of great value for the states of consciousness, behavior, and individual actions.

#### 1.1.4. Attention in the Field of Economy

The struggle to engage attention with economic intentions began in the third decade of the 19th century with the emergence of the mass press [63], although attention as an object of study of economic science began with the change in society introduced by computers in the late sixties of the 20th century [64].

Attention became an object of study when Herbert Simon [65] identified information as a future consumer good that competes for the attention of individuals. Attention becomes a commodity with the evolution of technological society. Simon pointed out “a need to allocate that attention—as a scarce good—efficiently among the overabundance of information sources that might consume it” [65]. The contest for the attention and action of the consumer on the inputs or received stimuli transformed the ways to access the user, since attention is a finite and limited human capacity and resource [45].

However, in the study of the attention economy, the changes go beyond the Information Society. Authors such as Franck argue that “information is the still-physical aspect of the trans-physical economy of attention” and what is fundamental about attention is the essence of being conscious [66]. Because of the massification of the internet and mobile devices, the digitization of life has been the perfect scenario for the attention economy due to the increasing number of information stimuli that people can receive [67,68]. Besides this, the digital universe includes the possibility of accurately measuring the attention of users (or consumers). 

Study of the attention economy has progressed in this way since the end of the 20th century, when Shapiro and Varian [67] identified web search engines as tools that allow classifying the information that people value, as will be carried out in a sophisticated way by monitoring algorithms [63]. In consequence, this introduces the personalization of the product [69], which implies a transformation regarding the massive and dispersed commercial and communication exchange [2].

As an economic resource, attention has been presented with four characteristics:It appears in a market in which products are sold and bought under the laws of supply and demand.It is scarce and has defined limits [16,17,45].It is finite [47].It implies a benefit increase to the most relevant actors; hence, the more attention engaged, the easier it is to grow it [69].

The previous context defines the attention time as a crucial variable [70] so that the cultural industries compete to capture the longest possible attention time from users. This is in social media, where a greater concentration of user or consumer attention is produced. This is the privileged place where knowledge, physical abilities, feelings, and attention are speculated [71,72].

Commercial competition for attention has been characterized as oligopolistic [73] and concentrated [66,74,75], since the technological elites and their platforms act as monopolies of attention. Indeed, the new media set aside the exchange of information for money, typical of the traditional economy, and made the capturing of attention the center of their business [74], as a commodity that is transformed into consumer data [76].

At the same time, the attention economy assumes that technological evolution, attractive interface designs, and sophisticated notification mechanisms allow the control of human attention. Social media activates attraction through emotional experiences [77], guarantees engagement with the platform and lengthens consumption time [47], and controls the moments of individual attention [78,79] of billions of users.

In the critical study of social media as control platforms, it has been highlighted that the information technology industry is the “most standardized and most centralized form of attentional control in human history (…) The attention economy incentivizes the design of technologies that grab our attention. In so doing, it privileges our impulses over our intentions” [80], something that transcends social media and that can be seen in media that is full of sensationalist bait, called clickbait.

From the economic field, the concept of attention is a rising value within cognitive capitalism [66]. Today, the maximum possible amount of information stimuli is received to compete for the cognitive resources of the individual. According to Franck [74], it is the most successful business model in the 21st century, thanks to media financed by advertising. Peirano [81] qualifies it as destructive because of the cognitive manipulation that it entails.

## 2. Materials and Methods

The article uses a theoretical and historical approach to explain the concept and functioning of human attention. This approach comprises three perspectives: neurophysiology, neuropsychology, and economics. For this purpose, the academic literature corresponding to these three disciplines in relation to attention has been critically reviewed, as is evidenced in the references list (with more than 100 items analyzed). Once the characteristics of the functioning of human attention have been explained, the authors proceed to relate them to the functioning of social media also from a theoretical perspective based on the existing literature. Thus, the authors construct and propose a theoretical model that describes the interactive design of social media based on the characteristics of attention previously explored.

The theoretical model described is subjected to a first empirical verification with the inclusion of two examples that show the utilization of the theoretical proposal. In addition, the application of the model to the examples linked to the use of social media is complemented by other similar studies which prove the validity of the theoretical model presented.

This article offers a theoretical starting point for future empirical research on the phenomenon of human attention in social media and the problems arising from the abuse of the currently dominant digital platforms.

## 3. Social Media, Interfaces, and the Control of Attention

The conceptualization of attention from economics, neurophysiology, and neuropsychology makes it possible to establish a relationship between the functioning of attention and the exercise of digital consumption. Therefore, some of the design and operational characteristics of social media can be identified to define them as a sophisticated form of control of individual attention [63,80]. The four mechanisms and their effects on human attention are summarized in Table 1:

### 3.1. Notifications as Systematic Impulses

Surprise, novelty [10], and repetition [82] are typical behaviors of the notification system of social media [83]. Notifications are systematic impulses that saturate the attentional network due to the information overabundance they represent. This saturation produces a state of overalert in the individual. Thus, the notification acts as a digital stimulus [67] for the alerting system [84], changing the neurophysiological state of the individual [23]. It acts as a distractor and leads to problems of self-control and self-discipline [85].

Likewise, the information received as sensory input in the form of personalized notification [65] directly impacts the emotional system of the individual, associated with the second and third network of the Attention System theory [23,25], determining the priority of actions [65]. The emotional intensity of social media notifications [86] works because it is linked to psychological characteristics such as social appreciation, self-image, acceptance, social comparison, and recognition [87,88]. By affecting the main psychological emotions, notifications also generate anxiety in individuals who use social media [89]. This feeds both the anxiety of knowing what is behind the notification itself and the desire to obtain it [90], which generates disappointment or even depression in its absence [91].

The above duality can be expressed in terms of the Posner framework [24] by locating the bottom-up and top-down attention mechanisms. In the first case (bottom-up), the stimulus stands out by its characteristics, and the user wants to know what the notification contains. In the second case (top-down), the individual expects the notification stimulus to be associated with the achievement or fulfillment of goals [43], such as recognition, acceptance, or personal self-image.

Lastly, notifications capture attention because they are repetitive but novel. Repetition [47] does not represent, in this case, a problem for attention, since it generates surprising new emotions every time [68]. The user may crave a notification, but it remains uncertain until taking the selective action of paying attention to it. Although most notifications are not relevant [92], they are designed to be noticed as new, unique, and changing [93]. One of the reasons for individuals to be “Always On” [94] is the novelty of the notification acting on the ARAS.

### 3.2. Social Media Messages and Posts

The structural functioning of social media apps is described as fundamental to promote addiction in users [77,95]. The messages or publications received through the platforms can be described as data of an audiovisual nature, which are easily sent and consumed due to their short, fast, dynamic, and changing structure. Their structure and diverse, surprising, and constant functioning generate attention breaks in the subject [10] and produce the phenomenon of attentional dispersion, depending on uninterrupted activity, affecting sustained attention.

The structure of the message has three characteristics that determine the high attention levels of the subject. In the first place, messages are associated with audiovisual language [96,97], pointing to emotions, the most impulsive framework of attention.

Secondly, messages are short and dynamic, which fit the scarce attention aspect [47,48]. For this reason, attention travels from one message to another within the platform and in each message, which functions as a new informational impulse renewing the cycle of attention. This is a characteristic that makes constant the sustaining of attention. Once the user has accessed the platform through the notification, an initial moment or capture of attention, the individual remains in the interface for a long time, which is the sustaining and permanence moment [47]. Then, the user receives a new notification when tries to leave the platform (a moment of completion), which leads to new infinite and automate service cycles. It is the competition for the market of user’s time [70] dominated by the oligopolies of attention [73].

Finally, to ensure attention and avoid fatigue by the repetition of the platform, social media constantly change the communication interface with the user. It is, thus, novel and surprising. It regains attention with renewed stimuli [10,82].

### 3.3. The Fear of Missing Out (FoMO)

The attention-grabbing power of social media [66] has generated the feeling that, if users do not constantly check their platforms, they will lose something important in their lives. Different studies identify the fear of missing out (FoMO) [98,99]. This is related to the impact of social media on user’s attention and its capture, even in addictive and psychologically problematic ways, which cause anxiety and stress.

FoMO can be understood from a neurophysiological perspective as the information flow and interaction that captures the attention coming from the activation of the filiation sensory mechanisms [10,47]. The relationship generated between the subject and social media, which is deeply dependent, produces psychological pathologies.

Simultaneously, attention also functions through involuntary electrochemical reactions that occur on received stimuli [100]. The FoMO is determined by the action of social media that feed the need for its use and that issues constant notifications maintaining and reinforcing user anxiety: “something is happening, and I might be missing it”.

### 3.4. Intermittent and Variable Rewards

The anxiety of receiving a signal of social approval determines the emotional functionality in terms of attention, of the intermittent and variable rewards [2]. Just as in a slot machine, the user inserts a coin, operates a button or a lever and craves a reward. The waiting time provokes a high level of uncertainty about the expected reward and, at the same time, generates a distance between the expected and received reward [101]. The uncertainty design is inherent to the interfaces and behavior of social media.

The communication interfaces design systems of social media, which act as the likes/rewards conjunction, construct reinforcements for behavioral stimuli, meaning promotion, and guidance for the actions of individuals. This action–reaction mechanism promoted by the system of likes and the intensely interactive design of social media converts the attention given into action or behavior of the individual and generates, with repetition and constant feedback, the systemic gratification itself of the platform, an addiction.

Addicted to the possible affirmative answer [102,103], users consume social media with the anxiety of reaching the jackpot [104,105], fueled by the values of individual recognition, selfishness, and popularity of the consumer society of the 21st century [74].

## 4. Confirmation of the Social Media Design Efficiency

The theoretical development can be empirically verified with two social media consumption case studies investigated in 2019 and 2020. The first research is summarized in the monitoring of the social media usage executed on a group of 25 university students (19–21 years old) in December 2019. It consisted of the observation and elaboration of weekly reports of the most common smartphone applications usage time. The study found that the average time spent on the mobile phone is 4 h and 26 min per day, of which 85% of the time was dedicated to social media [4]. In the study, complemented by the realization of four focus groups developed with the 25 participants, some of the theoretically recognized social media attentional effects were verified, such as FoMO, anxiety, attention dispersion, and addictive behavior.

As said by the participants regarding social media, “you always need to be connected… to know what is happening. It is like a vital necessity. In social media you feel as if you were part of something, if you leave, it is as if you stop being part of it”. This statement exemplifies a direct relationship with mechanisms such as FoMO or the anxiety of connecting to social media [98,99,102,103]. Another student mentioned that “we feel alone if we are outside the social media, I think that everyone is afraid of being nobody”, and thus assumed the psychological universe of the emotional stimuli of social media consumption [22,23]. Meanwhile, another participant pointed out that “the vitality, the lights, the colors, everything is striking. It attracts us, we have everything in that place, on our mobile and, especially on social media”. The last statement describes the alerting system—notifications—that attracts the attention of platform users [10,82,84]. At the same time, it confirms the consolidation of interaction and information in social media [66,75].

The consumption of the social media is identified as addictive [103] by young people: “sometimes you access for one thing and end up doing many others without realizing it, you lose the sense of time… we are addicted and we are becoming even more dependent”, one statement that coincides with the neurophysiological and neuropsychological attention types previously described concerning social media [47].

Similarly, some of the young people monitored stated that “sometimes you don’t even know what you see, but you feel the need to be there, on the screen, just scrolling”, which serves to show the operation of the message structure and the design of the platform [47,65], as well as the effectiveness of notifications to sustain prolonged attention for a long time on social media.

The second research was based on a survey carried out between February and May 2020 with 740 people in Spain (*n* = 740, with a mean age of 23.1 years). It showed that the average use of social media is 5.1 h per day, which is equivalent to almost 36 h per week. The survey was carried out among a population with different levels of education (22.8% had completed secondary school, 52.5% were undergraduates, 11.4% did vocational education, and 13.4% were postgraduates. They all declared to have access to the internet with a smartphone). The results are similar to other academic articles that show the long-time consumption of social media [106,107].

Furthermore, the results reveal that 55.8% of the sample considers that social media are addictive. Other participants consider social media as socialization tools (52.97%) or related to bullying (20.41%), as well as a source of social recognition (15.14%). Social media are also considered to generate feelings of saturation (14.46%) and have a high component of irrationality (5.14%).

The self-declaration data of the respondents show the activation of the attention processes defined in the theoretical perspective. Although these are preliminary investigations to measure and quantify attention, it is demonstrated that the design of social media is aimed at dominating the user’s attention, with satisfactory results for the platforms.

## 5. Conclusions

Attention can become a key element in understanding the consumption system of social media, despite its epistemological complexity. The article offers a theoretical approach to the concept of human attention using three different perspectives—neurophysiology, neuropsychology, and economy—which is a novel contribution. In the same way, the research connects the design of social media with its effects on human digital wellbeing. The article contributes to understanding the neurophysiological and neuropsychological basis of the concept, which helps to better structure the voluntary and involuntary organic functioning of attention economy.

Thanks to the theoretical construction of attention, the conceptual approach reveals the effects that social media consumption has on human attention. If attention is conceived as the gateway for the other processes of the human being to operate, the excessive consumption of social media is a threat that generates a series of repercussions for the physical and mental health. Social media—with its infinite and automated capacity of generation and reproduction of stimuli—condition the neurophysiological and neuropsychological systems and alter the behaviors of the subjects, both in their individuality and in their socio-affective development.

Sensory notification instruments work strategically on the emotions of the users and reproduce the cycle of capturing attention. The personalized mechanisms represent an electrochemical activation pattern within the neurophysiological and neuropsychological functions. Similarly, the effects on the massive capture of attention through platforms’ attractive designs—which act directly on the emotional system—determine the generation of a new profitable system of commodities that targets the capitalist exploitation of the cognitive and creative resources of users on a few platforms.

Some practical implications stem from the above, especially in terms of bringing two issues to the attention of citizens. On the one hand, it is necessary to raise awareness of the dangers of abuse in the consumption of social media for individuals, but above all thinking about the future of people’s digital welfare. On the other hand, it is fundamental to point out the economic oligopolies that control our attention and from which addictive technology is derived. This leads us to think about the need to establish self-regulation codes or good practices, as well as legislation that guarantees digital rights as human rights.

The theoretical approach of the article, however, has some limitations. The results of the cases presented to complement the explanation confirm the need to build longitudinal studies of social media consumption to gain a better understanding of the phenomenon. Likewise, it is important to develop a greater empirical approach, especially in a qualitative way, to explore in more detail the problem of the excessive use of social media and its problematic effects on human attention and digital wellbeing.

Although the study presents in a novel way the union of three perspectives, the concept of attention is much more complex and can be explored from other points of view not contemplated in this research.

The study also leads to future research. As mentioned before, more longitudinal analyses are needed. Although valuable data can be obtained from surveys or from qualitative approaches, research on social media; mobile devices; and their use, problems, or users’ motivations requires new methodological approaches to better understand the level, time, and attention paid to specific content. These methodological advances, such as information collected through brain monitoring with encephalography or eye-tracking techniques, will help us to understand the effects that social media use has on people’s attention. However, there is a need to qualitatively investigate the underlaying reasons for the intensive use of social media. Finally, another important line of future research is to confirm the public health impacts of the problematic use and concentration of human attention on social media.

## Figures and Tables

**Table 1 healthcare-08-00497-t001:** Mechanisms of social media operation and effects on human attention.

Mechanism	Stimulus	Affectation	Effect
Notifications	Systematic and overabundant auditory, visual, and sensory impulses	Attentional network saturation	Overalert state Anxiety
Messages and posts	Audiovisual frequent, short, simple, diverse, interactive, and dynamic data	Constant attention breaks	Attentional dispersion
Fear of missing out (FoMO) Oligopolistic control environment of attention	Apprehensive graphic and functional environment, and flow of information and interaction from the selection of topics and people of interest to the user to create a filiation relationship	Dependency	Anxiety Stress
Likes and rewards	Social acceptance stimulus that reinforces the behavioral character resulting from the user’s interaction or publication	Assimilation and normalization of repeated consumption behaviors	Addiction
**Source:** Authors’ elaboration			

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
