# Peer review of "The Struggle for Human Attention: Between the Abuse of Social Media and Digital Wellbeing"

_healthcare, 2020, doi:10.3390/healthcare8040497_

Round 1

Reviewer 1 Report

Thank you for presenting this research, there are some very interesting concepts addressed, and a linkage between social media use, attention and behavioural economics is a fruitful avenue of research.

However, there are substantial difficulties in the presentation/expression for an English language audience. In particular, there is too much focus on "historical" research that can be better expressed by synthesising a contemporary understanding of attention.

The main problem, however, is with regards to the strength of the research and conclusions that can be drawn from the data presented.

You describe "experiments" conducted, however this is very much not an experimental design - there is no variable manipulated, nor comparison of groups. It cannot be described as "quasi-experimental" either, as there are no groups or categories described from within the dataset. This makes the research largely descriptive, and while the second study shows a decent sample size, the conclusions are very limited and do not extend the available literature on problematic social media use.

In addition, a key problem is that there is no mechanism by which attention is observed in the methodology.

As such, I cannot recommend this article for publication.

Having said this, I would very much encourage continuing down this research pathway and developing stronger research in this area.

Author Response

The authors are grateful for the comments of the evaluator, who recognises the interest of the concepts presented. The ideas related to attention, as well as their link to social media, represent in fact a field of future research.

The text, indeed, is presented as a theoretical discussion. The main objective is to theoretically develop how the attention debate has been addressed.

In the first part, the authors use a historical perspective to, subsequently, develop the concept of attention from the perspective of neurophysiology, neuropsychology and economics. This is the focus of the article and how it can be derived from a theoretical approach a practical application for the study of the functioning of social media, with its specific problems of attention.

Then, two studies are presented as examples and as a complement to the theoretical explanation. The two studies presented at the end, in a non-canonical way within the universe of scientific publication, do not represent the focus of the study. The central idea of the article is the theoretical discussion, as indicated in one of the options in the call for papers for the special issue of the Journal.

The addition of two examples of studies at the end of the article is intended to show that the theoretical development of attention has an empirical connotation and that it is important to investigate the concept further, since the results of the different studies cited support the results of these two investigations presented, as well as the theoretical discussion.

However, the authors have rewritten the second part of the abstract, with the intention of giving more clarity on the role of the two examples presented. Likewise, the conclusions have been adapted giving also more relevance to the theoretical approach presented.

Reviewer 2 Report

The article is poor in terms of methodology. It should be specifically pointed out in a section of its own. On the other hand, the results should be presented in a graphic way to facilitate understanding.

Finally, the conclusions are not supported by the results except in the last paragraph. They are too open and inconclusive.

The presentation of a discussion that relates the theoretical framework to the analysis of the results would be appreciated.

The bibliography section is extensive and very complete. Congratulations

Author Response

The authors appreciate the suggestions and comments received. The goal of the text is indeed to produce a theoretical article, as one of the options mentioned in the call for papers of the special issue. Thus, the methodology focuses on the descriptive development of the theories of attention and the article provides a historical overview of their analysis. In the same way, emphasis is placed on the construction of basic relationships between theory and the (also theoretical) functioning of the social media.

The authors think that the results of the two final studies (which serve as an example of the relationship presented) cannot be more than descriptive to give rise to extensive research on them.

We believe that the relationship established in section 3 advances the discussion between the theoretical perspective of attention and the development of the interactive construction of social media. In fact, there is a theoretical-conceptual construction that includes many of the authors and concepts discussed in sections 1 and 2 of the text.

The authors are specially grateful for the last comment of the Reviewer, in which he/she congratulates the extensive bibliographic work, as this refers to the same central point: the text is presented as a theoretical reflection between attention (and its functioning, parts 1 and 2) and the construction of the interactive design of social media (and its functioning, part 3).

The last part of the article reflects the empirical nature of this theoretical universe and opens the door to fundamental questions that the first two Reviewers raise: how attention can be measured, and which are the main studies that reflect an addictive consumption in social media.

With the aim of clarifying these questions, the authors have rewritten the second part of the abstract and some of the conclusions have been revised to give greater relevance to the proposed theoretical approach.

Reviewer 3 Report

I would like to thank the efforts by the authors of the manuscript and congratulate them on the work. The work is ambitious and the results confirm the most of the hypotheses and the relevance and potential of the work is therefore recognized, but this Reviewer considers that several changes are needed to the manuscript is publishable. Some are of content, and other formal.

Remember that no references should appear in the manuscript (see reference 22).

In the conclusions section I suggest adding a paragraph on study limitations and one dedicated to commenting on future lines of research and practical implications. It should highlight the novelty of the study in comparison with previous empirical evidence of the same or similar topic.

Overall it's a good work that could be improved adding greater conceptual clarity in the ideas presented.
It assumes a good work of study conforms to the objectives and establishes a good starting point for further research on the topic and its practical implications.

Best wishes for Authors.

Author Response

The authors are glad to receive the Reviewer's feedback and welcome his/her suggestions, especially for improving the conclusions of the paper.

The authors have reviewed the Journal's style guide and have adapted the entire manuscript, as well as they have revised the references. In the same way, a thorough revision of the text has also been made to improve both language and structure issues to try to gain more clarity in the ideas presented.

In particular, the conclusions have been improved by adding the novelty and the limitations of the study, the practical implications and the future lines of research as rightly indicated by the Reviewer. Many thanks to the Reviewer.

Reviewer 4 Report

The paper theoretically analysed how human attention works from the perspectives of neurophysiology, neuropsychology and economy, and it demonstrated how the design of social media works to dominate human attention and cause addiction. It also used 2 case studies as empirical evidence of the theoretical statements. The topic is very interesting and relatively new. It illustrates how the design of social media works to capture human attention and affect emotion and behaviour.

However, several aspects of the paper need to be improved.

1. It seems that the paper uses neuroscience as an equivalent term to neurophysiology. For example, the Abstract section says "...from three disciplines: neurophysiology, neuropsychology, and economics.", but the introduction section says "from the neurosciences, neuropsychology, and the attention economy." The use of terms need to be consistent in the paper.

2. The paper used lots of quotes from existing literature, but some quotes need more explanation to facilitate understanding. For example, line 233-234: "information is the still-physical aspect of 233 the trans-physical economy of attention” - it's hard to understand without elaboration. 

3. Line 247-252 listed four characteristics of attention as an economic resource which can be related to neurophysiological or neuropsychological concepts, as said in line 246, but you didn't explain how they relate to neurophysiological or neuropsychological concepts, and specifically which concepts?

4. In Section 3, based on the conceptualization of attention from economics, neurosciences, and neuropsychology, the relationship between functioning of attention and characteristics of social media is established. In Table 1, can you specify which mechanism and effect is related to/from which discipline? Also, in Table 1, if "Attentional dispersion" is included as an effect, why didn't you also include "sustained attention" as an effect? Moreover, the other 3 mechanisms are design characteristics of social media, but the 3rd one (Section 3.3) is a psychological phenomenon from the impact of social media on users, so this needs to adjusted. Perhaps it can be included in both 3.1 and 3.2. Acturally when you talked about anxiety and "disappointment or even depression in its absence" in 3.1, it has already overlapped with "the fear of missing out".

5. In Section 4, line 371-372: "some of the theoretically recognized social media 371 attentional effects" - please specify which. Also, more information is needed about participants of the 2 case studies. For example, in the 1st study, did the 25 students participated in all the 4 focus groups? Their age? In the 2nd study, participants' educational level and access to internet and mobile devices? These can be confounding factors. How was the survey developed? Reliability test?

6. Some citations are lacking. For example, line 80:"...it is discovered that it is a limited capacity"; line 83:"...such as Watson and Skinner..."; line 341-342: "the filiation sensory mechanisms."

7. Some editing/language errors. For example, line 193-195 - grammatically it is not a sentence; line 246 - "-:" at the end of the sentence; line 321: "-." at the end of the sentence; line 348 - "1.4".

8. Conclusion can include some suggestions on practices, e.g., multiple factors and guidance to be considered for social media design to avoid addiction, in addition to the suggestions on future research.

Author Response

Many thanks to the Reviewer for his/her comments, the authors appreciate them for the improvement of their work.

  1. The authors have reviewed the use of specific terms and have preferred the term "neurophysiology" as the most appropriate. To avoid confusion, the use of the term "neuroscience" has been eliminated where it was not appropriate. In this way, as the Reviewer suggests, the authors believe that the article has gained consistency.
  2. The authors have reviewed and regrouped the paragraphs to better explain the quotations. For example, in the case of this comment, the quotation is supported by other arguments to explain and continue the idea expressed by the author. In any case, the corpus of quotes chosen for the article seeks to present the bibliographical universe where the topic is circumscribed. Therefore, some citations only intend to mention and leave open the possibility of questions or future theoretical deepening on the subject.
  3. Although the list of four characteristics is related to the neuroscientific development presented, the mention in this line asks for the corresponding clarification. The authors decided to eliminate it to avoid confusion. In this way, the relationship with neurophysiological and neuropsychological concepts will be better explained in the table presented and its explanation.
  4. The problem in specifying it is that all the mechanisms and effects have a neuropsychological and neurophysiological implication. For this reason, the authors prefer to present the open relation between the three perspectives used. In the next paragraph of the text, we develop some specific implications in the neuropsychological and neurophysiological areas.

Sustained attention is not an effect because this attention moment or function has been interrupted by the messages and the dynamics that follow them. Regarding the comment, we would like to clarify it, with the explanation in the 3.2 section of the text: “Its structure and diverse, surprising, and constant functioning generate attention breaks in the subject [9] and produces the phenomenon of attentional dispersion, depending on its uninterrupted activity, affecting the sustained attention.”

FoMO is a communicative concept to describe the user's feeling of constantly checking his/her platforms so as not to miss out on something important. It is not an effect because the term captures a media phenomenon. We understand the Reviewer’s comment because in the description paragraph (3.3) we present it as an effect, and it is probably confusing. We correct the text in this part to clarify this impression.

In the last comment of point 4, we would like to explain that stress and anxiety are general in all the mechanisms, as they are neuropsychological effects of social media consumption. Specifically, in FoMO these two characteristics are explicitly reflected. Thus, when we present in 3.1. section the mention of anxiety and "disappointment or even depression in its absence" it is because they are included in this point and we refer to them from a neurophysiological perspective. In the same way, the mechanism of social media operates as a whole. This is a theoretical distinction for analysis purposes.

  1. The authors have specified the effects to provide greater clarity of ideas. Similarly, following the Reviewer's suggestions, information has been added in the cases presented as examples at the end of the text, to avoid the misperception noted. The examples, which serve as a complement to the theoretical explanation, are presented in a brief and descriptive manner with the intention of continuing to deepen the practical derivatives of this theoretical research in the future.
  2. All quotations, citations and references have been checked to avoid confusion, as pointed out by the Reviewer.
  3. The authors have carried out a thorough review of the stylistic and linguistic aspects of the article to gain consistency and clarity.
  4. The conclusions section, in line with several comments received, has been rewritten to add the lines of future research and its practical implications, as well as the limitations of the study. However, the authors consider that they cannot suggest specific practices at this point of the research, beyond some recommendations included in the conclusions.

Round 2

Reviewer 1 Report

This is a strong submission. I appreciate now that the manuscript was intended to provide a historical context in order to frame conclusions about the relation between attention and social media design and engagement. Rather than as a context for the research per se. I also am impressed by the improvements made to English language expression.

The work raises some very interesting conclusions and will help to contextualise future research work.

Author Response

The authors are very grateful for your second review, as your comments have helped us to make the article clearer. Therefore, the authors would like to thank you especially for the time and effort put into improving our manuscript with your suggestions.

We look forward to continue progressing from the conclusions of the article.

Likewise, the authors wish you the best for your research.

Reviewer 2 Report

I strongly recommend merging the first parts. In MDPI journals "Introduction" is not a part to make an initial and short statement. You should include all theoretical background in this first part. After this minor change, it will be closer to MDPI way of publication.

About "Materials and Methods" I recommend the authors write a different epigraph showing shortly the way they have developed its bibliographical research.

I appreciate your job. Conclusions are now better and much deeper which changes all the paper.

I also appreciate the job of improving English writing. As an Spanish speaker and writer I'm not qualified to judge your English style, but first version had big mistakes.

Author Response

The authors are grateful for your valuable contributions to the improvement of the manuscript. Thus, to get closer to the MDPI way of publication, we have followed your suggestion by merging the first parts of the article. The corresponding numbers of the sections have been modified.

As for your second observation, the authors have included the "Materials and Methods" section, although the very theoretical nature of the manuscript (which is different from that of a purely bibliographic article) means that this section must necessarily be brief. In this sense, we hope that the intention of proposing a theoretical model for the future empirical analysis of the phenomenon is correctly expressed in this section.

We are very glad and grateful for the time and effort you have devoted to improving our manuscript with your suggestions, and we wish you the best for your research.